# Structured Expert Routing with Multi-View Task Priors for Offline Meta-Reinforcement Learning

**Yisen Zhao** [1] **Peixi Peng** [1] [2] **Xinyu Hu** [1] **Cong Li** [1] **Zhan Su** [1] **Zhuojian Li** [1]

## Abstract

Offline meta-reinforcement learning requires agents to generalize to unseen tasks from fixed datasets, yet existing sequence-based and MoE-based methods rely on implicit or token-level routing signals that fail to capture task-level structure. We propose the **Task-Guided Router (TGR)**, a structured expert-routing framework that explicitly models inter-task relationships via multi-view task representations that combine semantic descriptors, behavioral summaries, and latent dynamics features. Using structure-guided routing, TGR assigns experts based on global task compatibility rather than local trajectory fragments, enabling stable specialization and effective knowledge transfer across tasks. Extensive experiments on continuous-control benchmarks demonstrate that TGR consistently outperforms state-of-the-art offline meta-RL methods in few-shot generalization, particularly under sparse data and heterogeneous dynamics. Our results highlight the importance of task-level priors for robust offline meta-reinforcement learning.

## 1. Introduction

Reinforcement learning (RL) (Kaelbling et al., 1996; Li, 2017) has achieved strong performance in specialized domains, yet generalizing from *seen* training tasks to *unseen* test tasks—without additional interaction or retraining—remains a fundamental challenge (Beck et al., 2025). Meta-reinforcement learning (meta-RL) tackles this challenge by learning a *learning-to-learn* capability from a distribution of tasks (Wang et al., 2016). In the more restrictive setting of *offline meta-RL* (Prudencio et al., 2023), agent must acquire such adaptation capability solely from fixed,

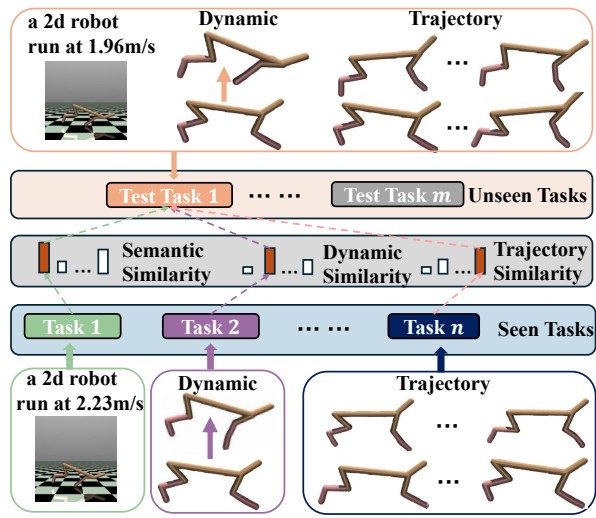

*Figure 1.* By leveraging **semantic, dynamic, and trajectory** priors to establish connections between unseen and seen tasks, our method matches tasks based on similarity across these three information sources, guiding the router to select experts from relevant seen tasks and thereby facilitating effective generalization to unseen environments.

pre-collected datasets, which is critical for applications where online exploration is expensive, unsafe, or infeasible.

The key challenge in meta-RL is how to adapt a policy model trained on seen tasks to unseen tasks. To address the gap between task distributions, Prompt-DT (Xu et al., 2022) and Meta-DT (Wang et al., 2024) extend the Decision Transformer (DT) (Chen et al., 2021), which generates actions by conditioning on past trajectories as task-specific prompts. Despite these advances, these methods (Brandfonbrener et al., 2022; Wu et al., 2023; Badrinath et al., 2023; Bhargava et al., 2023) treat all seen tasks uniformly and ignore that an unseen task may be more closely related to some seen tasks than to others. For example, in the cheetah-vel setting, the training (seen) tasks may include low-, medium-, and high-speed categories. When evaluating an unseen task with a high-speed objective, the agent may need to move slowly at the beginning to maintain balance and stabilize its posture. A prompt-based model can then incorrectly infer that the task goal corresponds to a low-speed task,

---

[1]School of Electronic and Computer Engineering, Peking University [2]Pengcheng Laboratory. Correspondence to: Peixi Peng <pxpeng@pku.edu.cn >.

*Proceedings of the 43rd International Conference on Machine Learning*, Seoul, South Korea. PMLR 306, 2026. Copyright 2026 by the author(s).

leading to a mismatch with the true objective and degraded performance on the unseen task.

To build the relationship between seen and unseen tasks explicit, we introduce three types of task-shared nodes for each task: (1) Semantic information, which links unseen and seen tasks by matching the similarity of their textual descriptions; (2) Behavioral trajectory information, which matches the state patterns in offline trajectories from seen tasks to the states observed in the test-time unseen task to infer the motion phase of the unseen task; and (3) Latent dynamics information, which provides a physical bridge between seen and unseen tasks by capturing shared dynamics properties. Trajectory nodes encode task-level behavioral regularities (capturing *what* states are visited), whereas dynamics nodes encode timestep-dependent transition signals (capturing *how* the environment evolves under actions).

To enable knowledge transfer to *unseen* tasks, we design **TGR** with a mixture-of-experts (MoE) (Shazeer et al., 2017; Zhou et al., 2022) backbone and a routing mechanism grounded in the three types of task-shared nodes. MoE architectures have been introduced in multi-task RL to mitigate negative transfer caused by dense parameter sharing, as exemplified by M3DT (Kong et al., 2025). M3DT primarily relies on task-specific prompts and performs well in in-distribution settings. However, M3DT explicitly injects task priors into routing to guide expert selection under out-of-distribution (OOD) (Mao et al., 2024) tasks. Different with existing works, TGR selects experts by summing two routing components: a *content score* and a *global score*. The content score captures fine-grained compatibility by measuring the relationship between the policy model's local hidden states and each expert. In contrast, the global score incorporates coarse-grained task priors by measuring the relationship between global task information (derived from the task-shared nodes) and each expert. By combining these two scores, TGR performs expert selection at complementary scales, facilitating information sharing and knowledge transfer from seen tasks to unseen tasks.

**Our main contributions are summarized as follows.**

- We propose an explicit way to relate seen training tasks to unseen test tasks by leveraging shared structure across three complementary task priors: task semantics, behavioral trajectories, and latent dynamics.

- We introduce the *Task-Guided Router (TGR)*, a structured expert-routing framework that realizes this principle through a unified multi-view task representation space and routes experts by matching tasks via seen–unseen similarity.

- Through extensive experiments on continuous-control benchmarks, we show that TGR mitigates negative

transfer and improves few-shot generalization to unseen tasks under varying data quality conditions.

## 2. Related Work

### 2.1. Meta-Reinforcement Learning

Existing meta-RL approaches generally fall into two categories: *optimization-based* methods, which learn a policy initialization that can be adapted with a small number of gradient updates (Finn et al., 2017; Gupta et al., 2018), and *context-based* methods, which infer latent task representations from offline trajectories to condition the policy, such as PEARL (Rakelly et al., 2019), VariBAD (Zintgraf et al., 2019), Meta-DT (Wang et al., 2024), and related extensions (Fakoor et al., 2019; Zhang et al., 2021; Yuan & Lu, 2022; Gao et al., 2023; Li et al., 2024).

While effective under moderate task variation, these methods typically do not explicitly model relationships between seen tasks and unseen tasks. Thus, knowledge transfer in offline meta-RL remains largely implicit, which can limit robustness under out-of-distribution task shifts. In contrast, our work addresses this gap by leveraging explicit task relationships to guide generalization.

### 2.2. Decision Transformer in Multi-task RL

The Decision Transformer (Chen et al., 2021) first applies Transformer (Vaswani et al., 2017) architectures to offline RL, modeling trajectories as sequences of return, state, and action tokens to predict actions autoregressively. To extend this to multi-task and meta-learning scenarios, variants such as Generalization-DT (Furuta et al., 2021), Prompt-DT (Xu et al., 2022), Harmony-DT (Hu et al., 2024), CoPDT (Xue et al., 2026), Prompt Tuning DT (Rietz et al., 2025)inject task-specific information (e.g., short trajectory segments) as prefixes to guide generation.

However, these Transformer-based variants primarily rely on short trajectory segments as implicit proxies for task identity. While such prefix-based conditioning can capture local behavioral context, it provides limited information about the underlying task structure or dynamics. As a result, these methods struggle to generalize when test tasks exhibit distributional shifts that are not well represented by the observed trajectory segments.

### 2.3. Mixture-of-Experts in RL and Sequence Models

Mixture-of-Experts (MoE) (Shazeer et al., 2017) architectures improve scalability by activating sparse subsets of parameters through learned routing and have recently been adopted in RL to mitigate the limitations of fully shared models, as exemplified by M3DT (Kong et al., 2025). However, existing MoE-based RL approaches rely on largely

unstructured routing mechanisms, treating tasks as independent instances or depending exclusively on data-driven correlations. Consequently, they fail to capture the underlying relationships among tasks, which constrains their effectiveness in scenarios requiring robust generalization across diverse task distributions.

## 3. Preliminaries

### 3.1. Offline Meta-RL with Seen and Unseen Tasks

We consider offline meta-RL over a task distribution $p(\mathcal{T})$, where each task $\mathcal{T}_i \sim p(\mathcal{T})$ is modeled as a Markov Decision Process (MDP) with task-specific dynamics and reward functions. In this setting, the agent is trained on a fixed dataset $\mathcal{D}$ that contains trajectories collected from *seen* tasks and does not involve any interaction with the environment during training.

At meta-test time, the agent is evaluated on *unseen* tasks drawn from the same action space, potentially under distributional shifts in state distribution, latent dynamics or reward structure. The objective is to learn a policy $\pi_\theta$ from $\mathcal{D}$ that generalizes effectively to these unseen tasks under fixed and limited data.

### 3.2. Prompt-DT as a Policy Backbone

We adopt the Prompt Decision Transformer (Prompt-DT) (Xu et al., 2022) as a sequence-based policy backbone for offline meta-RL. For a task $\mathcal{T}_i$, Prompt-DT prepends a short prompt trajectory segment

$$\tau_i^* = \{(\hat{R}_1^*, s_1^*, a_1^*), \ldots, (\hat{R}_{K^*}^*, s_{K^*}^*, a_{K^*}^*)\}, \quad (1)$$

where $K^*$ denotes the prompt length, $s_k^*$ and $a_k^*$ are the state and action at step $k$ of the prompt, and $\hat{R}_k^*$ is the corresponding return-to-go token. This prompt is prepended to the task trajectory to provide task-specific conditioning.

Given a context window of length $K$, the input token sequence at timestep $t$ is then constructed as

$$\tau_{i,t}^{\text{input}} = [\tau_i^*, \hat{R}_{t-K+1}, s_{t-K+1}, a_{t-K+1}, \ldots, \hat{R}_t, s_t], \quad (2)$$

where $(s_t, a_t)$ denotes the state-action pair at timestep $t$, and $\hat{R}_t$ is the return-to-go computed from timestep $t$ onward.

These prompts are significantly shorter than full trajectories ($K^* \ll K$) and provide sufficient task-specific information for task identification without requiring full demonstrations.

### 3.3. Mixture-of-Experts in Reinforcement Learning

Mixture-of-Experts (MoE) (Shazeer et al., 2017) architectures aim to improve model capacity and scalability by activating only a sparse subset of expert networks for each input. A typical MoE layer consists of a set of experts $\{f_e\}_{e=1}^E$ and

a routing function $g(\cdot)$ that maps an input representation to a probability distribution over experts. Given an input $x$, the output of an MoE layer is computed as

$$y = \sum_{e \in \mathcal{E}(x)} g_k(x) \, f_e(x), \quad (3)$$

where $\mathcal{E}(x)$ denotes the selected top-$k$ experts according to the routing scores.

Recent work (Kong et al., 2025) integrates MoE mechanisms into sequence-based RL models to facilitate partial specialization across diverse tasks, which enables structured specialization and improves scalability when training on large-scale multi-task datasets.

### 3.4. MoE Routing for Meta-RL

However, the existing MoE routing strategy in RL operates at the *token level* or *trajectory-fragment level*, where routing decisions are primarily conditioned on local hidden representations. Such designs are effective in standard multi-task settings, as the routing function can leverage rich in-distribution trajectory patterns to activate appropriate experts. In meta-RL scenarios, however, the routing problem becomes fundamentally more challenging due to the shift from *seen* training tasks to *unseen* test tasks. In this setting, early trajectory segments or short context windows may not provide sufficient information to reliably infer unseen task identity or underlying structure.

Importantly, this limitation does not arise from the MoE formulation itself, but rather from the absence of explicit task-level structure in the routing inputs. This insight motivates the development of routing mechanisms that enhance hidden-state-based signals with additional task-aware information, thereby enabling more stable and interpretable expert selection in offline meta-RL.

## 4. Methodology

We propose **Task-Guided Router (TGR)**, a structured expert routing framework for offline meta-RL. TGR introduces explicit task-level structure to guide expert specialization and routing, enabling effective knowledge transfer from seen training tasks to unseen test tasks.

Our method builds upon Prompt-DT as a sequence-based policy backbone and augments it with heterogeneous task representations. Each task is represented through 3 complementary node types capturing semantic information, behavioral trajectories, and latent dynamics, respectively.

Routing decisions are made by aggregating information over these task representations, allowing expert selection to be informed by global task relationships observed during meta-training rather than local trajectory fragments.

This structured design promotes stable expert specialization and improves generalization under distributional shift. We next describe the construction of task representation nodes (Sec. 4.1) and the Task-Guided Router for expert routing (Sec. 4.2).

### 4.1. Task Representation Construction

To enable structured expert routing in offline meta-RL, we construct heterogeneous *task representations* that explicitly capture relationships among tasks observed during meta-training. Instead of treating tasks as independent instances, these representations provide a unified space in which tasks can be compared and related from multiple complementary perspectives, facilitating more coherent reasoning over task similarities. For each task $\mathcal{T}_i$, we instantiate three types of task nodes, each encoding a separate aspect of task identity. These representations are later used as inputs to the router to compute expert weights (Sec. 4.2).

**Semantic Task Nodes.** Semantic task nodes encode high-level task information that is invariant to trajectory-level noise. Given task metadata $m_i$ for task $\mathcal{T}_i$, we first extract a language embedding using a pretrained language model $\phi_{\mathrm{LM}}(\cdot)$ and then transform it with an $n$-layer MLP $f_{\mathrm{MLP}}(\cdot)$ to obtain the semantic node representation:

$$z_i^{\mathrm{sem}} = f_{\mathrm{MLP}}\big(\phi_{\mathrm{LM}}(m_i)\big), \qquad (4)$$

where $z_i^{\mathrm{sem}}$ serves as a stable prior over task similarity, even when trajectory observations are limited or noisy. Notably, we compute $z_i^{\mathrm{sem}}$ once per task and reuse it throughout training, so it does not introduce additional per-iteration computation. In Sec 4.2, we use $z_i^{\mathrm{sem}}$ to augment the Prompt-DT input and to compute routing scores.

**Behavioral Trajectory Nodes.** Instead of encoding entire trajectories, we construct behavioral trajectory nodes using only state observations. This choice decouples task representation from action distributions and policy-specific biases. Since the offline dataset are sampled from unknown distribution, the resulting state pool can exhibit severe distribution imbalance. To address this issue, we perform structure-aware state clustering to extract balanced and behaviorally coherent prototypes.

Specifically, we apply Structure Information Partitioning (SIP) (Zeng et al., 2025), which partitions the state similarity graph by minimizing structural information loss. Compared to distance-based clustering methods, SIP produces partitions that are less sensitive to uneven state densities and sampling noise, leading to more stable behavioral representations across heterogeneous offline datasets.

Let $\mathcal{D}_i = \{\tau_{i,j}\}_{j=1}^N$ denote the offline dataset of task $\mathcal{T}_i$, where each trajectory $\tau_{i,j} = \{(s_{i,j,t}, a_{i,j,t}, r_{i,j,t})\}_{t=1}^T$ con-

sists of state, action, and reward tuples. We randomly sample $M$ trajectories from $\mathcal{D}_i$ and construct a task-level state pool by collecting the states from the sampled trajectories (Zeng et al., 2025):

$$\mathcal{S}_i = \bigcup_{j=1}^M \{s_{i,j,t}\}_{t=1}^T, \qquad (5)$$

Instead of directly averaging states, which is sensitive to data imbalance and fails to capture behavioral structure, we construct a state similarity graph using a Gaussian kernel $\sigma$:

$$[A_i]_{pq} = \exp\Big(-\frac{\|s_p - s_q\|^2}{2\sigma^2}\Big),$$
$$s_p, s_q \in \mathcal{S}_i, \quad [A_i]_{pp} = 0. \qquad (6)$$

Setting the diagonal to zero removes trivial self-loops, ensuring that the graph reflects inter-state relational structure rather than state visitation frequency.

Given the similarity graph $A_i$, we apply Structure Information Partitioning (SIP) (Zeng et al., 2025) to partition the states into $K$ structurally coherent clusters:

$$\mathcal{C}_i = \mathrm{SIP\_Clusters}(A_i, K), \quad \mathcal{C}_i = \{\mathcal{S}_{i,k}\}_{k=1}^K. \qquad (7)$$

We set $K$ equal to the number of experts $E$, such that each cluster functions as a structural prototype guiding expert specialization.

Finally, each behavioral trajectory node is represented by the centroid of its corresponding cluster:

$$z_{i,k}^{\mathrm{traj}} = \frac{1}{|\mathcal{S}_{i,k}|} \sum_{s \in \mathcal{S}_{i,k}} s, \quad k = 1, \dots, K. \qquad (8)$$

$$z_i^{\mathrm{traj}} = \mathrm{Concat}\Big(z_{i,1}^{\mathrm{traj}}, z_{i,2}^{\mathrm{traj}}, \dots, z_{i,K}^{\mathrm{traj}}\Big), \qquad (9)$$

where $|\mathcal{S}_{i,k}|$ denotes the number of states in cluster $\mathcal{S}_{i,k}$.

**Latent Dynamics Nodes.** Latent dynamics nodes encode task-specific environment dynamics in a compact latent form, computed over a fixed context window of length $H$.

For task $\mathcal{T}_i$, we encode a context window of length $H$ using a pretrained RNN dynamics encoder:

$$z_{i,t}^{\mathrm{dyn}} = \mathrm{RNN}(s_{i,t-H+1}, a_{i,t-H+1}, r_{i,t-H+1}, \dots, s_{i,t}), \qquad (10)$$

where $s_{i,t}$ is the current state at timestep $t$. We include actions and rewards in the context to better capture task-specific dynamics. In Sec 4.2, we use $z_{i,t}^{\mathrm{dyn}}$ to augment the Prompt-DT input and to compute routing scores.

**Trajectory vs. Dynamics Nodes.** Behavioral trajectory nodes and latent dynamics nodes capture complementary

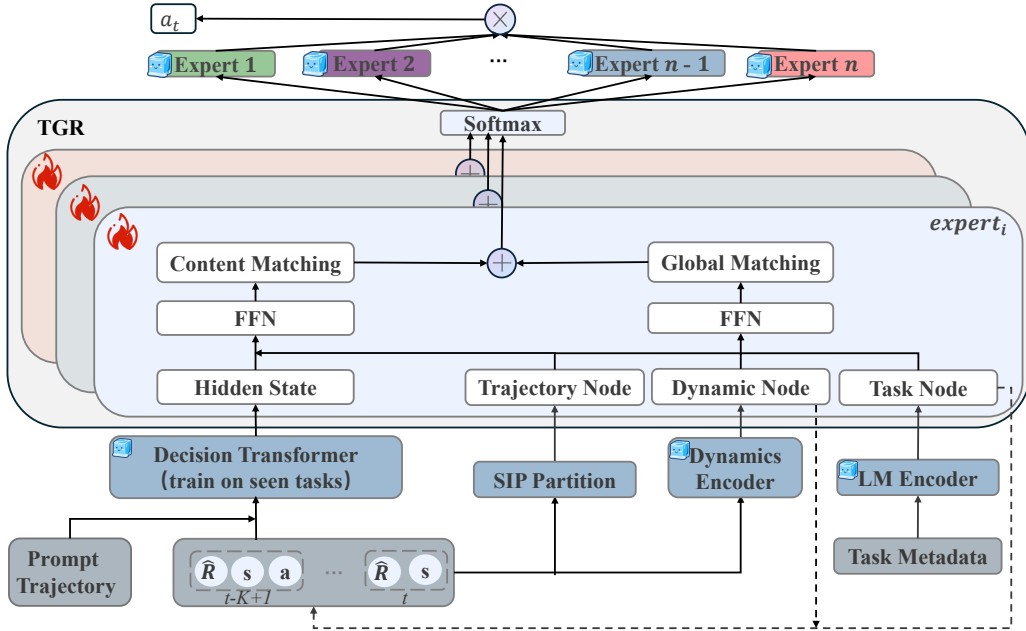

*Figure 2.* Overview of the proposed Task-Guided Router (TGR). TGR constructs heterogeneous task representations composed of semantic task nodes, trajectory nodes, and dynamics nodes. Given the hidden state from the Prompt-DT backbone and task-level representations, TGR computes node-wise routing scores and aggregates them into structured expert routing weights. The weighted combination of expert policies produces the final action prediction, enabling robust generalization from seen tasks to unseen tasks.

but distinct task signals. Trajectory nodes are task-level summaries computed by clustering state observations from the offline dataset; they characterize *what* regions of the state space are visited under the logged behavior, without relying on actions or temporal ordering. In contrast, dynamics nodes are timestep-dependent latent features extracted from short $(s, a, r)$ context windows; they characterize *how* the environment evolves and what outcomes follow actions. We therefore use $z_i^{\text{traj}}$ as a stable behavioral prior and $z_{i,t}^{\text{dyn}}$ as a context-conditioned signal for routing and for augmenting the Prompt-DT input.

**Complementarity of Node Types.** Each node type captures a distinct and complementary aspect of task identity. Semantic nodes provide high-level task descriptors that anchor routing decisions at the task level; trajectory nodes encode behavior patterns from offline data, enabling data-driven characterization of task execution; and dynamics nodes represent latent environment properties that facilitate structural knowledge transfer across tasks. By integrating these heterogeneous representations into a unified representation space, TGR enables structured reasoning about task relationships and supports robust generalization under distributional shift.

### 4.2. Task-Guided Router

As illustrated in Figure 2, given the constructed task representations, we introduce the *Task-Guided Router (TGR)* to

perform structured expert routing for offline meta-RL. The router determines how much each expert should contribute to the final action prediction by explicitly leveraging task relationships encoded by different node types. This design facilitates generalization from *seen tasks* to *unseen tasks* by decomposing routing into multiple interpretable signals.

TGR takes the Prompt-DT hidden representation $h_{i,t}$ and the task representations $\{z_i^{\text{sem}}, z_i^{\text{traj}}, z_{i,t}^{\text{dyn}}\}$ as inputs . For clarity, we use $\psi_{i,t}^{\text{input}}$ to denote the Prompt-DT input augmented with task-level signals.

**Training Pipeline.** Before training the router, we adopt a staged procedure to obtain the task-level signals and expert policies used by TGR. First, we pretrain a dynamics model on offline sequences using an RNN encoder that produces $z_{i,t}^{\text{dyn}}$ from the most recent $H$ steps, together with a multilayer MLP decoder that predicts the next-step state and reward. Second, we freeze the dynamics encoder and use it to compute $z_{i,t}^{\text{dyn}}$; in parallel, we use a pretrained language model (Liu et al., 2019) (kept frozen) to compute semantic embeddings $z_i^{\text{sem}}$ from task metadata. Third, we augment the Prompt-DT input with these task-level signals and train the policy backbone to obtain hidden representations $h_{i,t}$. Fourth, we partition training tasks into expert groups using task-specific gradient information from the pretrained Prompt-DT together with semantic similarity, train expert policies $\{\pi_e\}_{e=1}^{E}$, and finally train TGR to predict routing

weights while keeping the backbone, encoders, and experts fixed. More detailed information about training pipeline can be fund in Appendix D.

Since the Prompt-DT input already includes a prompt trajectory $\tau_i^*$, we do not additionally append the trajectory-level representation $z_i^{\text{traj}}$ to $\psi_{i,t}^{\text{input}}$. Concretely, we define the augmented input sequence as

$$\psi_{i,t}^{\text{input}} = [\tau_i^*, z_i^{\text{sem}}, \hat{R}_{t-K+1}, z_{i,t-K+1}^{\text{dyn}}, s_{t-K+1},$$
$$a_{t-K+1}, \ldots, z_i^{\text{sem}}, \hat{R}_t, z_{i,t}^{\text{dyn}}, s_t], \qquad (11)$$

and compute the corresponding hidden state by

$$h_{i,t} = \text{Prompt-DT}(\psi_{i,t}^{\text{input}}). \qquad (12)$$

**Routing Objective** Let $\{\pi_e\}_{e=1}^E$ denote a set of expert policies, each producing an action prediction given the current trajectory context. At timestep $t$, the final action is computed as a weighted combination of expert outputs:

$$\hat{a}_t = \sum_{e=1}^E w_{i,t}^{(e)} \pi_e(\psi_{i,t}^{\text{input}}), \qquad (13)$$

where $w_{i,t}^{(e)}$ is the routing weight assigned to expert $e$. The routing weights are obtained by normalizing the unnormalized node-wise routing scores:

$$w_{i,t}^{(e)} = \frac{\exp(s_{i,t}^{(e)})}{\sum_{e'=1}^E \exp(s_{i,t}^{(e')})}, \quad \sum_{e=1}^E w_{i,t}^{(e)} = 1. \qquad (14)$$

The challenge is to infer meaningful routing weights for unseen tasks with limited trajectory data.

**Node-wise Routing Scores** TGR computes expert routing scores by aggregating contributions from different task node types. Specifically, for each expert $e$, the unnormalized routing score is:

$$s_{i,t}^{(e)} = s_{i,t}^{(e,\text{content})} + s_{i,t}^{(e,\text{global})}. \qquad (15)$$

This decomposition reflects two complementary sources of routing signals that are critical in offline meta-RL. The content term models instance-level compatibility between the current trajectory context and each expert, enabling adaptive routing based on partial observations from the target task. In contrast, the global term encodes task-level priors derived from task representations, capturing global structural relationships among seen tasks and providing a stable bias for expert selection under distributional shift.

By combining these two components through additive aggregation, TGR integrates local context-aware evidence with global task-aware priors. This design is particularly important for unseen tasks, where early trajectory segments may be noisy or insufficient to reliably infer task identity, and routing decisions benefit from additional task-level structural signals learned during meta-training.

*Content Routing Score* The content score captures similarity between the current hidden state and each expert:

$$s_{i,t}^{(e,\text{content})} = Q_{i,t} \cdot (K_e^{\text{content}})^\top, \qquad (16)$$

$$Q_{i,t} = f_{\text{query}}([h_{i,t}, z_i^{\text{sem}}, z_{i,t}^{\text{dyn}}, z_i^{\text{traj}}]), \qquad (17)$$

where $Q_{i,t}$ is the query computed from the hidden state $h_{i,t}$, semantic embedding $z_i^{\text{sem}}$, dynamics representation $z_{i,t}^{\text{dyn}}$, and trajectory embedding $z_i^{\text{traj}}$, and $K_e^{\text{content}}$ is the learnable key vector for expert $e$. This query construction corresponds to the multi-source fusion module shown in Figure 2.

*Global Routing Score* The global bias score reflects the suitability of each expert based on task-level signals:

$$s_{i,t}^{(e,\text{global})} = G_{i,t} \cdot (K_e^{\text{global}})^\top, \qquad (18)$$

$$G_{i,t} = f_{\text{global}}([z_i^{\text{sem}}, z_{i,t}^{\text{dyn}}, z_i^{\text{traj}}]), \qquad (19)$$

where $G_{i,t}$ is a global query computed from task-level embeddings, and $K_e^{\text{global}}$ is a learnable global key vector for expert $e$. This global term provides a task-level prior that can transfer across tasks sharing similar dynamics, even when semantics or trajectories differ.

## 5. Experiments

We conduct a comprehensive empirical evaluation to assess the effectiveness of the proposed Task-Guided Router (TGR) in offline meta-RL. Our experiments are designed to answer the following questions: (i) whether TGR improves few-shot generalization to unseen tasks compared to strong offline meta-RL baselines; (ii) how individual components of TGR contribute to its performance and; (iii) whether TGR scales to standard multi-task reinforcement learning settings. (iv) whether the performance of TGR comes from improved task structure modeling rather than increased parameter capacity. Morever,Appendix C shows the performance of different baselines after adding metadata.

Unless otherwise specified, all results are averaged over five random seeds, and shaded regions or error bars indicate one standard deviation. Moreover, we follow the experimental setups of Meta-DT (Wang et al., 2024) to ensure fair and comparable evaluation across methods.

**Environments.** All environments, task distributions, and offline datasets are configured consistently with prior work, and all reported results adhere to the same training and evaluation protocols.

We evaluate all methods on two widely used benchmarks. The first benchmark is **Point-Robot** (Gao et al., 2023),

*Table 1.* Few-shot test returns on unseen tasks under different dataset qualities. The last column reports the average normalized score (avg norm score), computed by min–max normalizing each environment's mean return and averaging across environments.

*(a)* Medium-quality datasets

| Method | Point-Robot | Cheetah-Vel | Ant-Dir | Hopper-Param | Walker-Param | avg norm score |
|---|---|---|---|---|---|---|
| Prompt-DT | -12.58 ± 0.27 | -135.89 ± 19.91 | 287.23 ± 29.51 | 309.90 ± 5.74 | 357.30 ± 18.34 | 47.95 ± 3.62 |
| Generalized DT | -14.97 ± 0.43 | -123.47 ± 6.88 | 268.53 ± 12.20 | 320.77 ± 12.82 | 368.07 ± 13.79 | 47.75 ± 1.52 |
| CORRO | -14.51 ± 1.65 | -154.12 ± 28.83 | 246.75 ± 49.33 | 332.37 ± 12.90 | 368.02 ± 41.77 | 43.38 ± 5.72 |
| CSRO | -16.39 ± 2.95 | -111.83 ± 9.16 | 251.66 ± 30.01 | 332.84 ± 11.23 | 388.08 ± 24.92 | 48.87 ± 3.70 |
| M3DT | -14.67 ± 0.19 | -107.19 ± 3.97 | 237.47 ± 39.35 | 345.49 ± 3.26 | 349.45 ± 4.05 | 49.76 ±1.73 |
| Meta-DT | -10.18 ± 0.18 | -99.28 ± 3.96 | 412.00 ± 11.53 | 348.20 ± 3.21 | **405.12 ± 11.11** | 64.89 ± 0.95 |
| **TGR (Ours)** | **-9.77 ± 0.23** | **-92.64 ± 6.56** | **417.53 ± 39.57** | **370.52 ± 6.54** | 392.45 ± 8.29 | **67.01 ± 1.98** |

*(b)* Mixed-quality datasets

| Method | Point-Robot | Cheetah-Vel | Ant-Dir | Hopper-Param | Walker-Param | avg norm score |
|---|---|---|---|---|---|---|
| Prompt-DT | -15.31 ± 1.52 | -91.34 ± 14.87 | 869.58 ± 22.64 | 320.76 ± 10.58 | 391.24 ± 20.28 | 60.34 ± 3.08 |
| Generalized DT | -15.10 ± 0.52 | -86.52 ± 10.62 | 511.97 ± 23.86 | 338.36 ± 14.07 | 394.32 ± 19.78 | 55.03 ± 2.14 |
| CORRO | -10.38 ± 1.29 | -81.59 ± 37.17 | 255.49 ± 32.64 | 335.12 ± 26.73 | 330.54 ± 14.07 | 52.76 ± 6.48 |
| CSRO | -18.14 ± 3.75 | -110.55 ± 12.01 | 330.15 ± 33.77 | 351.96 ± 13.61 | 334.34 ± 28.57 | 42.49 ± 4.49 |
| M3DT | -15.16 ± 0.06 | -95.35 ± 1.86 | 645.21 ± 87.90 | 360.49 ± 1.99 | 403.31 ± 11.81 | 57.40 ± 1.85 |
| Meta-DT | -8.39 ± 0.28 | -79.90 ± 5.69 | **908.48 ± 20.73** | 358.05 ± 10.69 | **470.36 ± 12.85** | 74.60 ± 1.26 |
| **TGR (Ours)** | **-7.01 ± 0.03** | **-70.58 ± 5.39** | 874.63 ± 19.77 | **376.34 ± 11.65** | 456.51 ± 24.48 | **77.03 ± 1.46** |

*(c)* Expert-quality datasets

| Method | Point-Robot | Cheetah-Vel | Ant-Dir | Hopper-Param | Walker-Param | avg norm score |
|---|---|---|---|---|---|---|
| Prompt-DT | -7.99 ± 0.46 | -133.78 ± 18.24 | 678.07 ± 68.74 | 393.79 ± 11.44 | 449.15 ± 37.53 | 61.99 ± 3.72 |
| Generalized DT | -12.99 ± 0.34 | -62.95 ± 3.42 | 613.59 ± 49.22 | 358.56 ± 11.75 | 421.96 ± 40.70 | 65.01 ± 2.07 |
| CORRO | -7.76 ± 0.18 | -111.47 ± 36.97 | 381.42 ± 13.83 | 338.17 ± 47.36 | 352.02 ± 44.99 | 53.90 ± 6.70 |
| CSRO | -19.42 ± 2.10 | -129.00 ± 24.24 | 417.37 ± 39.70 | 358.29 ± 16.25 | 336.89 ± 16.71 | 40.23 ± 4.72 |
| M3DT | -10.21 ± 0.24 | -123.47 ± 9.50 | 709.85 ± 49.54 | 383.79 ± 12.45 | 430.88 ± 11.23 | 61.00 ± 2.00 |
| Meta-DT | -6.90 ± 0.11 | -52.42 ± 8.11 | 961.27 ± 18.07 | 383.51 ± 8.99 | 437.79 ± 18.21 | 81.44 ± 1.62 |
| **TGR (Ours)** | **-5.69 ± 0.21** | **-35.77 ± 2.69** | **985.67 ± 25.85** | **404.73 ± 15.45** | **451.49 ± 32.70** | **87.31 ± 1.61** |

a 2D navigation environment with parametric variations across tasks. The second benchmark comprises continuous-control tasks from **MuJoCo** (Todorov et al., 2012), including *Cheetah-Vel*, *Ant-Dir*, *Hopper-Param*, and *Walker-Param*. In each domain, the task distribution is split into a training set $\mathcal{T}_{\text{train}}$ and a disjoint test set $\mathcal{T}_{\text{test}}$ to evaluate generalization to unseen tasks.

**Baselines.** We compare TGR against representative offline meta-RL methods from two paradigms. Decision Transformer-based approaches include **Prompt-DT** (Xu et al., 2022), **Generalized DT** (Furuta et al., 2021), and **Meta-DT** (Wang et al., 2024). We additionally include temporal-difference (TD) learning baselines, namely **CORRO** (Yuan & Lu, 2022) and **CSRO** (Gao et al., 2023). All baseline results are taken from the Meta-DT paper.

### 5.1. Few-shot Generalization

Few-shot generalization is the primary evaluation criterion in offline meta-RL, where an agent must adapt to previously unseen tasks using only a small amount of offline context data and without additional environment interaction. Following prior work (Wang et al., 2024; Gao et al., 2023), we adopt a standardized few-shot evaluation protocol.

At test time, each method is provided with a fixed number of trajectories collected from the target task, which are used to infer task-specific information, either as prompt prefixes or as context windows for DT-based approaches. To ensure a fair comparison, all methods are given access to the same number of context trajectories during evaluation.

Table 1 reports the converged test returns on unseen tasks using three quality levels: *Medium*, *Mixed*, and *Expert* following established protocols (Levine et al., 2020; Yuan & Lu, 2022).

**Average normalized score.** For each method and environment, we compute a normalized score from the mean return

*Table 2.* Ablation results on Cheetah-Vel and Hopper-Param.

| Method | Cheetah-Vel | Hopper-Param |
|---|---|---|
| Prompt-DT | -135.89 ± 19.91 | 309.90±5.74 |
| + Dyn | -112.51 ± 3.24 | 340.87 ± 1.72 |
| + Dyn + Sem | -110.95 ± 6.68 | 352.19 ± 1.84 |
| MoE (M3DT) | -107.19 ± 3.97 | 345.49 ± 3.26 |
| + Dyn | -106.49 ± 5.02 | 348.05 ± 2.04 |
| + Dyn + Sem | -102.88 ± 7.01 | 364.59 ± 9.74 |
| + Dyn + Sem + Traj | **-92.64 ± 6.56** | **370.52 ± 6.54** |

*Table 3.* Effect of routing score design.

| Routing Strategy | Cheetah-Vel | Hopper-Param |
|---|---|---|
| Content | -110.48 ± 5.14 | 351.24 ± 1.43 |
| Global | -99.29 ± 2.99 | 362.45 ± 1.78 |
| Content+Global | **-92.64 ± 6.56** | **370.52 ± 6.54** |

*Table 4.* Ablation on input information sources in TGR.

| Model Variant | Cheetah-Vel | Hopper-Param |
|---|---|---|
| Full TGR | **-92.64 ± 6.56** | **370.52 ± 6.54** |
| w/o Sem | -105.75 ± 4.49 | 367.24 ± 2.05 |
| w/o Traj | -102.88 ± 7.01 | 364.59 ± 9.74 |
| w/o Dyn | -102.31 ± 4.70 | 360.18 ± 1.13 |
| w/o Dyn&Traj | -120.18 ± 9.25 | 355.90 ± 2.89 |
| w/o Dyn&Sem | -110.26 ± 4.87 | 356.10 ± 1.72 |
| w/o Sem&Traj | -106.49 ± 5.02 | 348.05 ± 2.04 |

*Table 5.* Multi-Task RL Performance on 160 tasks

| Method | Score |
|---|---|
| M3DT | 78.21 ± 0.47 |
| TGR (Ours) | **79.90 ± 0.64** |

$R$ (Kong et al., 2025) as

$$\text{score} = 100 \cdot \frac{R - R_{\min}}{R_{\max} - R_{\min}}, \tag{20}$$

and clip it to $[0, 100]$. We then average the scores across the five environments to obtain avg norm score. The detailed information about avg norm score can be fund in Appedix A.

Across all evaluated environments, TGR achieves superior performance in the majority of settings compared to strong baselines, including Meta-DT, Prompt-DT, Generalized DT and TD-based approaches. These results suggest that explicitly modeling inter-task relationships with task-level structure, combined with structured expert routing, effectively mitigates gradient interference and enhances generalization in offline meta-RL.

### 5.2. Ablation Study

To quantify the contribution of each design choice in TGR, we conduct ablation studies on two representative benchmark environments: *Cheetah-Vel* and *Hopper-Param*. We report the results in Tables 2–4.

**Progressive ablations.** We perform progressive ablations that incrementally add task priors and routing structure to minimal baselines. As shown in Table 2, we study two tracks: (i) Prompt-DT → Prompt-DT+Dyn → Prompt-DT+Dyn+Sem; and (ii) MoE (M3DT) → MoE+Dyn → MoE+Dyn+Sem → MoE+Dyn+Sem+Traj (full TGR).

**Routing score design.** To isolate the effect of the routing objective, we ablate the score decomposition in Table 3 by evaluating (a) routing with only the Content score, (b) routing with only the Global score, and (c) routing with their combination (Content+Global). This comparison directly

tests whether task-level priors (Global) and local token-level compatibility (Content) provide complementary signals for expert selection.

**Input information sources.** Finally, Table 4 evaluates the sensitivity of TGR to each information source by removing one or two priors from the full model, including w/o Sem, w/o Traj, w/o Dyn, and their pairwise removals. These ablations assess the necessity of semantic, trajectory, and dynamics information as inputs to the routing mechanism under the same backbone and training setup.

Overall, across Tables 2–4, progressively adding task priors consistently improves performance, and the full TGR achieves the best returns, indicating that each module contributes positively rather than acting as redundant capacity. Moreover, Content+Global routing outperforms either score alone (Table 3), supporting the complementarity between task-level priors and token-level compatibility. Finally, removing any single prior (or especially pairs) degrades results (Table 4), validating the advantage of integrating semantic, trajectory, and dynamics information for expert selection.

### 5.3. Multi-Task Reinforcement Learning

Finally, we evaluate TGR in standard multi-task reinforcement learning (MTRL) settings. We conduct experiments on a 160-task benchmark composed of tasks from MetaWorld and DMControl, together with the Ant-Dir and Cheetah-Vel task families. As a representative baseline, we compare TGR against M3DT (Kong et al., 2025), a recent state-of-the-art multi-task Decision Transformer that employs mixture-of-experts with token-level routing.

Overall, Table 5 shows that TGR achieves strong in-distribution performance across the full multi-task training set, and that explicitly modeling inter-task relationships pro-

*Table 6.* Performance comparison under similar parameter scales.

| Method | Params | Cheetah-Vel | Hopper-Param |
|---|---|---|---|
| Meta-DT (Large) | ∼34M | -97.46 ± 5.28 | 350.47 ± 3.62 |
| M3DT | ∼33M | -107.19 ± 3.97 | 345.49 ± 3.26 |
| **TGR (Ours)** | **∼33M** | **-92.64 ± 6.56** | **370.52 ± 6.54** |

vides a more effective inductive bias for MTRL than purely token-based MoE routing.

### 5.4. Fair Comparison on Training Budget and Parameter Scale

To verify that the performance gains of TGR do not simply arise from increased model capacity, we additionally compare TGR with scaled-up baseline models under matched parameter budgets. Specifically, we enlarge the Meta-DT backbone by increasing the number of Transformer layers, attention heads, and embedding dimensions, resulting in a model size comparable to TGR (∼33M parameters). Moreover, the total number of gradient update steps in each training stage is kept identical to those of the baselines to ensure a fair comparison.

As shown in Table 6, simply increasing the parameter scale of baseline models does not lead to consistent performance improvements. In contrast, TGR achieves superior results under comparable model size, indicating that its gains primarily arise from structured task priors and the proposed routing mechanism rather than increased capacity.

## 6. Conclusion

We propose Task-Guided Router (TGR), a task-structured expert routing method for offline meta-RL that leverages semantic, trajectory, and dynamics signals to enable expert specialization and transfer. TGR consistently improves few-shot generalization on continuous-control benchmarks, outperforming strong baselines and highlighting the value of explicit task-structure modeling.

Nevertheless, our current framework still has several limitations. In future work, we plan to further explore how to more tightly integrate multi-view task priors, and investigate how to maintain robust performance under more challenging out-of-distribution (OOD) task settings as well as offline meta-RL scenarios with image-based observations.

## Impact Statement

This paper presents work whose goal is to advance the field of Machine Learning. There are many potential societal consequences of our work, none which we feel must be specifically highlighted here.

## Acknowledgement

The study was funded by the Shenzhen Science and Technology Program (KQTD20240729102051063), the National Natural Science Foundation of China under contracts No. 62422602, No. 62372010, No. 62425101, No. 62332002, No. 62372010, No. 62206281, Key Laboratory Grants 241-HF-D05-01, and the major key project of the Peng Cheng Laboratory (PCL2021A13 and PCL2025A02). Computing support was provided by Pengcheng Cloudbrain.

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

# A. Experiment Details

In this section, we show details of evaluation environments over a variety of testbeds, as well as the offline dataset collection process conducted on these environments.

## A.1. Detailed Environments

Following the previous studies in offline meta-RL (Wang et al., 2024), we adopt two classical benchmarks:the 2D navigation environment (Gao et al., 2023),the multi-task MuJoCo control environment (Todorov et al., 2012). We evaluate all tested methods on the following environments as

- **Point-Robot:** A point agent navigates in a 2D plane to reach a goal position. The observation consists of the agent's 2D coordinates; the goal location is not included in the observation. The action space is $[-0.1, 0.1]^2$, where each dimension specifies the displacement along the horizontal and vertical axes, respectively. The reward is the negative Euclidean distance between the agent and the goal. Each episode starts from the fixed origin and terminates after a horizon of 20 steps. Tasks vary by the goal position, which is sampled uniformly from the unit square, inducing different reward functions across tasks.

- **Cheetah-Vel and Ant-Dir:** Multi-task MuJoCo continuous-control benchmarks where tasks differ only in the reward function. In **Cheetah-Vel**, a planar cheetah is required to run at a target speed along the positive $x$-axis; the reward decreases with the absolute deviation between the agent's current speed and the goal speed. In **Ant-Dir**, a 3D quadruped ant is trained to move along a target direction, and the reward is given by the cosine similarity between the agent's velocity and the goal direction. The goal speed for Cheetah-Vel is sampled uniformly from $\mathcal{U}[0.075, 3.0]$, and the goal direction for Ant-Dir is sampled uniformly from $\mathcal{U}[0, 2\pi]$. The maximum episode length is set to 200 steps.

- **Hopper-Param and Walker-Param:** Multi-task MuJoCo benchmarks in which tasks share the same objective but differ in transition dynamics. The agent controls a one-legged hopper or a two-legged walker and is encouraged to run as fast as possible. The reward is proportional to the forward velocity along the positive $x$-axis and is kept identical across tasks. Task variations are induced by randomizing physical parameters, including body mass, inertia, damping, and friction. As a result, the agent must move forward under changing environment dynamics. The maximum episode length is set to 200 steps for both benchmarks.

For the Point-Robot and MuJoCo environments, we sample 45 tasks for training and another 5 held-out tasks for testing. And Table 7 shows the $R_{\min}$ and $R_{\max}$ used for computing the avg norm score.

*Table 7.* Min/max ranges used for computing the avg norm score.

| Dataset quality | Point-Robot | Cheetah-Vel | Ant-Dir | Hopper-Param | Walker-Param |
|---|---|---|---|---|---|
| Medium | $[-20, 0]$ | $[-150, -30]$ | $[0, 500]$ | $[0, 500]$ | $[0, 500]$ |
| Mixed | $[-20, 0]$ | $[-150, -30]$ | $[0, 1000]$ | $[0, 500]$ | $[0, 500]$ |
| Expert | $[-20, 0]$ | $[-150, -30]$ | $[0, 1000]$ | $[0, 500]$ | $[0, 500]$ |

# B. The Details of Baselines

This section describes five representative baselines: three Decision Transformer (DT)-based methods, one temporal-difference (TD)-based method, and one diffusion-based method. We select these baselines to cover the three major categories of offline meta-RL approaches. In addition, since our method *TGR* is a DT-based method, we include multiple DT-based baselines for a more fine-grained comparison within this family. The baselines are summarized as follows:

- **Prompt-DT** (Xu et al., 2022) is a DT-based method that combines Transformer sequence modeling with prompting to enable few-shot adaptation in offline RL. It constructs a *trajectory prompt* from segments of a small set of demonstration trajectories, which encodes task-relevant information to condition policy generation. At test time, the method assumes access to a handful of expert demonstrations to form the prompt.

- **Generalized DT** (Furuta et al., 2021) is a DT-based approach that casts multi-task learning as a *hindsight information matching* (HIM) problem. The goal is to learn policies that generate the remainder of a trajectory so that it matches specified statistics of future information. It further instantiates offline multi-task state-marginal matching and imitation learning as two HIM objectives, and evaluates the resulting variants (Categorical DT and Bi-directional DT).

- **Meta-DT** (Wang et al., 2024) is a DT-based offline meta-RL method that conditions a causal Transformer policy on a learned task representation. It pretrains a context-aware world model with disentanglement to infer a compact task embedding, and injects this embedding as contextual information to guide task-oriented sequence generation. Meta-DT further constructs a self-guided trajectory prompt from history trajectories generated by the meta-policy, selecting the segment that yields the largest prediction error under the pretrained world model to capture task-specific information complementary to the learned representation.

- **CORRO** (Yuan & Lu, 2022) is a temporal-difference (TD)-based method that uses contrastive learning to obtain task representations robust to behavior-policy mismatch between training and testing. It formulates representation learning as mutual information maximization between the task variable and the learned representation, with the aim of removing behavior-policy-specific information. At test time, CORRO infers the task representation from a context trajectory collected by an arbitrary policy.

- **CSRO** (Gao et al., 2023) is a TD-based method that introduces a max–min mutual information representation learning objective to address the discrepancy between training contexts (from the behavior policy) and test-time contexts (from the exploration policy). At test time, CSRO explores a new task and collects a small amount of context data to infer the task representation.

## C. Utilization of task metadata across baselines.

To evaluate the impact of task metadata utilization in offline meta-RL, we further augment several representative baselines, including Prompt-DT, Meta-DT, and M3DT, with the same task metadata used in TGR. Specifically, we adopt an identical integration strategy to the construction of Semantic Task Nodes described in Section 4.1, where task metadata is concatenated with the input representation in a consistent manner across all methods.

The comparison results are shown in Table 8.

*Table 8.* Effect of incorporating task metadata into baseline methods.

| Model | HalfCheetah-Vel | Hopper-Param |
|---|---|---|
| Prompt-DT + Metadata | $-126.50 \pm 8.86$ | $348.60 \pm 13.86$ |
| Meta-DT + Metadata | $-97.93 \pm 4.66$ | $357.23 \pm 1.09$ |
| M3DT + Metadata | $-120.18 \pm 9.25$ | $355.90 \pm 2.89$ |
| **TGR (Ours)** | **$-92.64 \pm 6.56$** | **$370.52 \pm 6.54$** |

As shown in Table 8, incorporating task metadata into baseline methods leads to moderate performance improvements in some cases. However, TGR consistently achieves significantly better performance across all tasks. This indicates that the gains of TGR do not solely come from the usage of task metadata, but rather from its structured task-guided routing mechanism, which enables more effective utilization and specialization of task information.

## D. Implementation Details of TGR

We adopt a staged procedure to obtain the task-level signals and expert policies used by TGR. First, we pretrain a dynamics model on offline sequences using an RNN encoder that produces $z_{i,t}^{\mathrm{dyn}}$ from the most recent $H$ steps, together with a multi-layer MLP decoder that predicts the next-step state and reward. Second, we freeze the dynamics encoder and use it to compute $z_{i,t}^{\mathrm{dyn}}$; in parallel, we use a pretrained language model (Liu et al., 2019) (kept frozen) to compute semantic embeddings $z_i^{\mathrm{sem}}$ from task metadata. Third, we augment the Prompt-DT input with these task-level signals and train the policy backbone to obtain hidden representations $h_{i,t}$. Fourth, we partition training tasks into expert groups using task-specific gradient information from the pretrained Prompt-DT together with semantic similarity, train expert policies $\{\pi_e\}_{e=1}^E$, and finally train TGR to predict routing weights while keeping the backbone, encoders, and experts fixed.

**Dynamic Model.** We adopt lightweight architectures for the context-aware world model, including a context encoder, a reward decoder, and a state transition decoder. The context encoder consists of a GRU and a fully connected multilayer perceptron (MLP), both using ReLU activations. The GRU encodes an $h$-step history $(s_{t-h}, a_{t-h}, r_{t-h}, \ldots, s_{t-1}, a_{t-1}, r_{t-1}, s_t)$ into a 128-dimensional vector, which is then mapped by the MLP to a 16-dimensional embedding, i.e., the task representation $z$. The reward decoder is an MLP that takes $(s, a, s', z)$ as input and predicts the reward $r$ using two 128-dimensional hidden layers. Similarly, the state transition decoder is an MLP that takes $(s, a, r, z)$ as input and predicts the next state $s'$ using two 128-dimensional hidden layers.

**Language Model.** We use RoBERTa-base (`FacebookAI/roberta-base`) as the language model. RoBERTa is a Transformer encoder pretrained on a large-scale English corpus in a self-supervised manner using the masked language modeling (MLM) objective (Liu et al., 2019). Concretely, the pretraining procedure randomly masks a subset of input tokens and trains the model to recover the masked tokens conditioned on the full surrounding context, resulting in bidirectional contextual representations. The RoBERTa-base checkpoint we use is case-sensitive, i.e., it distinguishes between `english` and `English`.

**Task Metadata Template.** We standardize task metadata using a unified text template shared across all benchmarks. Each task is described by the same set of fields, including the task name, a short natural-language description, the environment family (e.g., MuJoCo or Point-Robot), and a structured specification of the goal (or target) state space. We then concatenate these fields into a single sentence-level prompt and feed it into RoBERTa to obtain a semantic embedding $z_i^{\text{sem}}$. This design provides a consistent input format for semantic encoding, reducing spurious variation caused by wording and length differences across tasks. As a result, semantic similarities become more comparable across tasks, which stabilizes routing based on $z_i^{\text{sem}}$ and improves expert specialization and transfer. It also makes the framework scalable and reproducible: adding a new task only requires filling the same metadata schema, without changing the model architecture or training procedure.

We implement the proposed **TGR** based on the official codebase of M3DT (Kong et al., 2025) and adopt most of its hyperparameter settings. Specifically, task representations $z$, returns-to-go $\hat{R}$, states $s$, and actions $a$ are first mapped to modality-specific linear embeddings. We then add the same positional and episodic timestep encoding to tokens from the same timestep. The resulting token sequence is fed into a GPT-style architecture, which predicts actions autoregressively using a causal self-attention mask.

**Expert Design.** Our proposed MoE architecture replaces the FFN with an MoE in each transformer block. This is based on what is common practice when adding MoEs to transformer architectures, but is by no means the only way to utilize MoEs.

**A Coarse-to-fine Task Selection.** We select task subsets using a two-stage clustering pipeline based on gradient, semantic text, and dynamic context features. First, we compute gradient-conflict features by elementwise multiplying each task gradient with the average gradient (after removing all-zero dimensions), and apply K-means ($K_c=3$), choosing the run that minimizes the variance of cluster sizes. Second, within each coarse cluster, we fuse $\ell_2$-normalized text and context features (text reduced to 16 dimensions via PCA; context kept at 16 dimensions) with weight $\alpha=0.4$, and fit a Gaussian mixture model to obtain up to $K_f=3$ fine-grained groups. The resulting fine clusters define task groups; we use these groups to construct expert and subtask subsets in subsequent experiments.

All experiment in this paper are run with 3 seeds. The specific model parameters and hyper-parameters utilized in our training process are outlined in Table 9 and Table 10.

*Table 9.* Hyper-parameters of TGR in our experiments.

| Parameter | Value |
| --- | --- |
| Number of MLP | 2 |
| Number of layers | 6 |
| Number of attention heads | 8 |
| Hidden dimension | 256 |
| Number of experts | 9 |
| Nonlinearity function | ReLU |
| Batch size | 16 |
| Prompt length $K$ | 20 |
| Dropout | 0.1 |
| Learning rate | $5.0 \times 10^{-5}$ |
| Optimizer | Adam |
| Context Horizon h | 4 |

*Table 10.* Hyperparameters of TGR on Point-Robot and MuJoCo domains with various datasets.

| Dataset | Hyperparameter | Point-Robot | Cheetah-Vel | Ant-Dir | Hopper-Param | Walker-Param |
| --- | --- | --- | --- | --- | --- | --- |
| | Backbone training rounds | 100000 | 100000 | 400000 | 100000 | 100000 |
| Medium | Expert training rounds | 100000 | 100000 | 200000 | 100000 | 100000 |
| | Router training rounds | 100000 | 100000 | 100000 | 100000 | 100000 |
| | Backbone training rounds | 300000 | 200000 | 400000 | 100000 | 100000 |
| Expert | Expert training rounds | 200000 | 200000 | 200000 | 100000 | 100000 |
| | Router training rounds | 100000 | 100000 | 100000 | 100000 | 100000 |
| | Backbone training rounds | 300000 | 200000 | 400000 | 100000 | 100000 |
| Mixed | Expert training rounds | 200000 | 200000 | 200000 | 100000 | 100000 |
| | Router training rounds | 100000 | 100000 | 100000 | 100000 | 100000 |

