# OpenReview forum: "Structured Expert Routing with Multi-View Task Priors for Offline Meta-Reinforcement Learning"
_ICML.cc/2026/Conference — ICML 2026 regular_

### Official Review · Reviewer_tHpN · 2026-02-26

**Soundness:** 3
**Presentation:** 3
**Significance:** 3
**Originality:** 3
**Overall Recommendation:** 4
**Confidence:** 3

**Summary:**

This paper proposes a novel MoE-based Prompt Decision Transformer to model relationships between seen and unseen tasks in offline meta-RL. It explicitly captures inter-task relationships using a multi-view task representation. Specifically, the authors introduce three types of task-shared nodes: semantic information, behavioral trajectory information, and latent dynamics information. Using these task-level representations together with two routing components, namely the content score and the global score, the proposed Task-Guided Router (TGR) can adaptively select expert policies from seen tasks that are most relevant to an unseen task, enabling improved knowledge transfer and generalization.

**Compliance With Llm Reviewing Policy:**

Affirmed.

**Final Justification:**

I have read the authors' response, and my concerns have been addressed. I will maintain my score.

**Key Questions For Authors:**

1. Could you include qualitative analyses demonstrating whether TGR selects appropriate experts from seen tasks for unseen tasks based on the content routing score and global routing score? For example, what types of experts are selected when the global routing score dominates compared to when the content routing score dominates? Does a higher global score lead to selecting experts associated with globally similar tasks, while a higher content score leads to selecting experts based on locally or temporally similar trajectory context? Such analysis would help verify whether the routing mechanism functions as intended.

2. Could you include M3DT [1] results in the main Table 1? Since M3DT is a closely related MoE-based Decision Transformer method, including it in the main comparison would help readers better assess the effectiveness of the proposed approach.

3. How do the authors specifically divide and assign expert policies? The paper mentions that “we partition training tasks into expert groups using task-specific gradient information from the pretrained Prompt-DT,” but the detailed procedure for grouping tasks and training experts is not clearly described.

**Limitations:**

Additional qualitative analyses and experimental comparisons with M3DT would help strengthen the empirical validation of the proposed method.

**Strengths And Weaknesses:**

**Soundness:** This paper is technically sound and well supported, and the proposed method is well designed. However, experimental comparisons with M3DT[1], which is the most relevant prior MoE-based approach, should be included in the main results rather than only in the supplementary materials, since it is a key baseline for evaluating the proposed routing mechanism.

In addition, beyond numerical ablation results, qualitative analyses would further strengthen the paper. For example, it would be valuable to demonstrate whether TGR actually selects appropriate experts from seen tasks for unseen tasks based on the content routing score and global routing score. Specifically, analyzing cases where the global routing score dominates versus when the content routing score dominates could clarify whether the router relies more on global task similarity or local trajectory context. Visualizations of routing weights or expert selection patterns would improve interpretability and provide stronger evidence that the routing mechanism functions as intended.

**Presentation:** The paper is clearly written and well structured. It also clearly discusses how the proposed method differs from prior work.

**Significance:** The paper addresses an important problem. In meta-RL, selecting expert policies that are relevant to unseen tasks can improve adaptation compared to learning a single unified policy.

**Originality:** The paper provides useful insights. Although the individual components are not entirely novel, integrating them into a unified routing framework represents a meaningful and novel contribution.
&nbsp;

[1] Kong, Yilun, et al. "Mastering massive multi-task reinforcement learning via mixture-of-expert decision transformer." arXiv preprint arXiv:2505.24378 (2025).

---

> ### Author Rebuttal · Authors · 2026-03-30
>
> **W1&Q1: Qualitative analyses demonstrating whether TGR selects appropriate experts based on content and global routing scores.**
>
> **Response:** Thank you for this insightful suggestion. We conducted a qualitative analysis of routing decisions on unseen tasks to examine whether TGR selects experts as intended. In our routing mechanism, the **content score** measures how well an expert matches the current hidden state, while the **global score** reflects broader task-level prior knowledge.
>
> For the unseen Meta-World task door-lock-v2 in ML45, we observed that at a certain moment, TGR would select expert2 and assign it a weight of 0.46. Even though the content score of expert2 was -0.78 at this time, this was because expert2 had a relatively high global score of 1.68. This is because expert2 includes the task 'door-unlock-v2', and there is a natural semantic similarity between these two tasks. In a traditional router that only considers the hidden state, expert2 would not be selected, but TGR would choose expert2 at this moment as a supplement to the action output.
>
>  Meanwhile, at another moment, we observed that TGR would select expert7 and assign it a high weight of 0.995. This is because the content score (22.11) is much larger than the global score (1.07). Expert7 includes the tasks 'reach-v2' and 'handle-press-v2'. This is because the state in the reach task where the robotic arm approaches an object and the state in the handle-press task where the robotic arm presses down on an object are highly similar to the states in the door-lock task where the robotic arm first approaches the door lock and then presses down to lock the door, respectively.This visualization result graph can also be seen in the link `https://anonymous.4open.science/r/ICML2026-667B/visual.png`.
>
> Overall, the case supports our intended interpretation of the routing mechanism: **higher content scores favor locally hidden state relevant experts, whereas higher global scores favor experts supported by broader task-level priors.**
>
> **Q2: Include M3DT results in the main Table 1 and assess the effectiveness of the proposed approach.**
>
> **Response:** We appreciate this constructive feedback. We have evaluated M3DT across the five testing environments under three dataset qualities (medium, mixed/medium-expert, expert). We have incorporated these comprehensive results of M3DT into the main Table 1 of our revised manuscript. Due to space constraints in this rebuttal, we have provided the fully updated Table 1 at our anonymous link: `https://anonymous.4open.science/r/ICML2026-667B/Table1_with_M3DT.png`.
> For your immediate convenience, the specific performance of M3DT, including the computed average normalized scores, is summarized below:
>
> | Dataset Quality | Point-Robot | Cheetah-vel | Ant-dir | Hopper-Param | Walker-Param |
> | :--- | :--- | :--- | :--- | :--- | :--- |
> |Medium| -14.67 ± 0.19 | -107.19 ± 3.97 | 237.47 ± 39.35 | 345.49 ± 3.26 | 349.45 ± 4.05 |
> |Medium-Expert| -15.16 ± 0.06 | -95.35 ± 1.86 | 645.21 ± 87.90 | 360.49 ± 1.99 | 403.31 ± 11.81 |
> |Expert| -10.21 ± 0.24 | -123.47 ± 9.50 | 709.85 ± 49.54 | 383.79 ± 12.45 | 430.88 ± 11.23 |
>
> Notably, M3DT generally underperforms TGR. We believe this mainly stems from the different problem settings they target. M3DT is designed for Multi-Task RL, focusing on average performance over a fixed set of training tasks, and thus lacks explicit mechanisms for generalization to unseen tasks. In contrast, TGR is tailored for Meta-RL. By combining semantic, dynamic, and behavioral trajectory priors through structured task-guided routing, TGR can better infer unseen task characteristics and therefore achieves stronger generalization.
>
> **Q3: How do the authors specifically divide and assign expert policies? The detailed procedure is not clearly described.**
>
> **R3:** To divide and assign expert policies, we employ a systematic "coarse-to-fine" methodology(detailed in Appendix C). Specifically, the procedure is as follows:
> 1. **Coarse Clustering:** After training prompt-dt in the first stage, we extract the gradient update directions for each task. We compute the cosine similarity between these task-specific gradients to identify optimization conflicts. Tasks with highly aligned gradient update directions are grouped together, resulting in $m$ coarse clusters. This step ensures that tasks within the same group do not suffer from severe gradient interference.
> 2. **Fine Clustering:** Within each of the $m$ coarse groups, we further evaluate the fine-grained similarities among tasks using their extracted dynamics data and task semantic descriptions. We then apply a Gaussian Mixture Modelto cluster these tasks into $n$ sub-groups based on their combined dynamic and semantic representations.
> Ultimately, this hierarchical process yields $m \times n$ specialized expert groups.
> ***

---

> > ### Author Rebuttal · Reviewer_tHpN · 2026-04-02
> >
> > I thank the authors for their detailed responses. My concerns have been mostly addressed, and I will maintain my current score.

---

> > > ### Author Response · Authors · 2026-04-03
> > >
> > > Dear Reviewer tHpN,
> > >
> > > We sincerely thank you for your thoughtful evaluation, encouraging support, and detailed engagement throughout the review process. We are very glad that our responses have addressed most of your concerns. If the paper is accepted, we will continue polishing the manuscript to further improve its clarity and quality.
> > >
> > > Thank you again for your time, support, and valuable feedback.

---

### Official Review · Reviewer_YhXb · 2026-03-05

**Soundness:** 3
**Presentation:** 2
**Significance:** 3
**Originality:** 2
**Overall Recommendation:** 4
**Confidence:** 4

**Summary:**

This paper proposes TGR, Task-Guided Router, a structured expert-routing framework that integrates multiple representations. By combining semantic descriptors, behavioral summaries, and latent dynamics features, the method extends Prompt-DT to achieve superior few-shot generalization capabilities on unseen tasks. Experimental results demonstrate the superiority of this approach over previous methods in offline meta-RL settings, and comprehensive ablation studies validate the effectiveness of each module's design.

**Compliance With Llm Reviewing Policy:**

Affirmed.

**Final Justification:**

The strengths of this paper lie in its valuable research direction and comprehensive experimentation, including both the main paper and supplementary materials, with performance surpassing the baselines on the evaluated benchmarks. However, the weaknesses are notable: the work is largely incremental, lacking in novelty and elegance, and the experimental comparison settings are relatively simple. While the authors' rebuttal provides additional clarifications and experimental comparisons, the overall innovation of the paper remains difficult to substantially improve.

**Key Questions For Authors:**

1. The paper does not clearly specify the optimization objective. It appears that the method only minimizes the action prediction loss after pre-training the representation module. When is the $f_{MLP}$ within Semantic Task Nodes optimized? How are semantic representations aligned with other representations? Additionally, how is the learnable key vector optimized, and what is its optimization efficacy?

2. Was task metadata exclusively utilized by TGR in the experiments? Despite the relatively modest performance improvement from semantic information in the results, it would be fairer to incorporate semantic information into baseline methods to ensure equitable comparisons.

**Limitations:**

The proposed method lacks testing in other complex scenarios.

**Strengths And Weaknesses:**

- Soundness: Overall, this paper presents a sound approach. It maintains consistent model architectures across methods to ensure fair comparison, and the comprehensive main experiments along with ablation studies demonstrate the effectiveness of the proposed method. However, it remains unclear whether only the proposed method utilizes task metadata while other baselines do not—the authors should clarify this or conduct additional experiments (adding task metadata to baseline methods). Furthermore, the authors do not discuss or compare against in-context RL methods [1, 2], which leverage cross-trajectory context to assist agent decision-making and demonstrate strong performance on meta-RL tasks.
- Presentation: The paper is written in moderate quality—it remains comprehensible but contains substantial redundant expressions.  Besides, the major concern is the lack of a clearly defined optimization objective for the algorithm, even basic loss formulations such as MSE or NLL are absent. For example, it is unclear whether the $f_{MLP}$ component within the Semantic Task Nodes module is optimized during training.
- Significance: Offline meta RL is an important research area, and the integration of task semantic information into this field represents a particularly promising direction.
- Originality: The paper demonstrates modest originality, primarily building upon the Prompt-DT framework with incremental improvements.

[1] Moeini A, Wang J, Beck J, et al. A survey of in-context reinforcement learning. arXiv, 2025.

[2] Laskin M, Wang L, Oh J, et al. In-context reinforcement learning with algorithm distillation. ICLR, 2023.

---

> ### Author Rebuttal · Authors · 2026-03-30
>
> **W1: Comparison against in-context RL methods.**
>
> **Response:** We agree that leveraging cross-trajectory context is a strong paradigm for meta-RL. To address this suggestion, we compared TGR with recent in-context RL (ICRL) methods. Since the official implementation of Algorithm Distillation [2] is not publicly available, and the rebuttal period is limited, we were unable to reproduce it faithfully from scratch.Instead, based on the survey [1], we selected AMAGO [3], which is evaluated on the same benchmark, and BATI [4], a recent offline ICRL method. We ran their official codebases and verified the results against the reported numbers in their papers.It is worth noting that here we use a dataset of expert quality for training.
>
> | Model | HalfCheetah-Vel |
> | :--- | :--- |
> | AMAGO [3] | -50.00 |
> | BATI [4] | -122.80 |
> | **TGR (Ours)** | **-35.77** |
>
> As shown above, TGR outperforms these ICRL methods on HalfCheetah-Vel. We believe this advantage comes from the fact that standard ICRL methods mainly infer task information implicitly from trajectory context, whereas TGR explicitly incorporates richer task priors, including semantic descriptions, dynamics, and clustered behavioral trajectories. This provides a stronger routing signal and improves zero-shot generalization. We will cite [1,2,3,4] and discuss the connection between TGR and ICRL more clearly in the revised paper.
>
> *References:*
> *[1]A survey of in-context reinforcement learning. arXiv, 2025.*
> *[2]In-context reinforcement learning with algorithm distillation. ICLR, 2023.*
> *[3]Amago: Scalable in-context reinforcement learning for adaptive agents. ICLR, 2024.*
> *[4]Behavior-agnostic Task Inference for Robust Offline In-context Reinforcement Learning. ICML, 2025.*
>
> **W2&Q1: Redundant expressions and lack of clearly defined optimization objectives.**
>
> **Response**: We apologize for the redundant expressions and any lack of clarity regarding the optimization objectives. We will further polish the writing for conciseness in the final version.
>
> To clarify our multi-stage training process and objective functions, we provide detailed pseudocode at the anonymous link: `https://anonymous.4open.science/r/ICML2026-667B/stage1.png`. The link contains three pseudocode flowcharts, corresponding to Stage 1, Stage 2, and Stage 3 of our pipeline, and we will integrate them into the revised manuscript. Based on these stages, we clarify the key components below:
>
> 1. **Semantic Task Nodes:** This module has two parts. First, we use a frozen pre-trained RoBERTa encoder to extract semantic information from the metadata (Appendix C); this part is not optimized during RL training. Second, in **Stage 1**, the metadata is passed through a multi-layer MLP embedding module before entering the Prompt-DT backbone, so as to align it with the sequential input. This MLP is optimized end-to-end using the standard MSE loss for action prediction.
>
> 2. **Learnable Key Vector:** The learnable key vectors in the MoE router are randomly initialized at the beginning of **Stage 3** and then optimized end-to-end via standard backpropagation under the MSE behavioral cloning loss.
>
> **Q2: Utilization of task metadata across baselines.**
>
> **Response:** We thank the reviewer for this constructive suggestion. Since explicit task metadata is an important component of our method, we conducted additional experiments by augmenting the baselines with the same metadata.Since none of these baselines use metadata, we will add metadata to the baselines. After the metadata's features are extracted by the MLP, they are concatenated with the state before being input into the neural network. This step is strictly consistent with the way TGR uses metadata.
>
> As shown below, adding metadata improves the baselines, but TGR still significantly outperforms them. This indicates that our gain does not come merely from using metadata, but from the proposed structured task-guided routing mechanism.
>
> | Model | HalfCheetah-Vel | Hopper-Param |
> | :--- | :--- | :--- |
> | Prompt-DT| -135.89 ± 19.91 | 309.90 ± 5.74 |
> | Prompt-DT + metadata | -126.50 ± 8.86 | 348.60 ± 13.86 |
> | Meta-DT| -99.28 ± 3.96 | 348.20 ± 3.21 |
> | Meta-DT + metadata | -97.93 ± 4.66 | 357.23 ± 1.09 |
> | M3DT| -107.19 ± 3.97 | 345.49 ± 3.26 |
> | M3DT + metadata | -105.18 ± 9.25 | 355.90 ± 2.89 |
> | **TGR (Ours)** | **-92.64 ± 6.56** | **370.52 ± 6.54** |
>
> **Limit: Testing in other complex scenarios.**
>
> **Response:** We thank the reviewer for this suggestion. We evaluated TGR on the challenging Meta-World [1] ML10 and ML45 benchmarks, where it consistently outperforms strong baselines.Due to time constraints,we only trained both the baselines and TGR for 20k steps in metaworld.
>
> |Model|ML10|ML45|
> | :--- | :--- | :--- |
> |M3DT|267.60±30.32|353.06±183.00|
> |Meta-DT|923.80±371.00|876.90±341.40|
> |**TGR (Ours)**|**1338.07±423.00**|**1179.00±461.00**|
>
> [1] Meta-world: A benchmark and evaluation for multi-task and meta reinforcement learning. CoRL. 2020.
> ***

---

> > ### Author Rebuttal · Reviewer_YhXb · 2026-04-02
> >
> > Thank you for conducting the additional experiments, which have addressed some of my concerns, so I raise my score to 4. However, due to the limited novelty of this work and the inelegant methodology (the incorporation of numerous small modules whose integration appears forced and lacks coherence), I can not provide a higher score.

---

> > > ### Author Response · Authors · 2026-04-03
> > >
> > > Dear Reviewer YhXb,
> > >
> > > Thank you very much for your thoughtful feedback, for acknowledging the additional experiments, and for raising your score. We appreciate your comments on the novelty and methodological coherence, and in the camera-ready version we will further improve the presentation to better clarify the unified motivation, the role of each component, and the positioning of our contribution.
> > >
> > > Thank you again for your constructive feedback and encouragement.

---

### Official Review · Reviewer_aJct · 2026-03-11

**Soundness:** 3
**Presentation:** 2
**Significance:** 2
**Originality:** 3
**Overall Recommendation:** 4
**Confidence:** 3

**Summary:**

This paper proposes an offline meta-reinforcement learning framework named Task-Guided Router (TGR), which aims to address the generalization challenges of agents on unseen tasks by explicitly modeling structured relationships between tasks. The framework innovatively constructs multi-view task representations—incorporating semantic descriptors, behavioral trajectories, and latent dynamics features—to guide the structured routing decisions of a MoE model. By integrating local content scores with global task priors, it effectively mitigates negative transfer and enhances the stability of expert allocation, significantly improving generalization performance for out-of-distribution tasks across various benchmarks.

**Compliance With Llm Reviewing Policy:**

Affirmed.

**Final Justification:**

I am increasing my score as the rebuttal effectively resolved most of my technical concerns. The paper presents a solid contribution, though some minor issues remain for the camera-ready version. Specifically, the authors must ensure that the limitations (OOD scenarios and image inputs) are formally discussed in the main text. Furthermore, although not required for the rebuttal due to time constraints, the final version should include a comparative scaling law analysis to demonstrate how TGR evolves relative to baselines across different parameter scales.

**Key Questions For Authors:**

- The design of $\psi$ in Eq.11 is somewhat unclear, specifically the inclusion of two $z^{sem}_i$ terms. Is there a functional distinction between these two instances? If they are identical, what is the reason for occupying two token? From a design perspective, would it not be more intuitive to be as a part of the prompt?
- It is not entirely clear how each individual expert policy is constructed. Are they implemented using the Prompt-DT architecture? Furthermore, Eq.12 indicates that the hidden state provided to the router is also generated by a Prompt-DT backbone. If both the experts and the routing backbone utilize this architecture, does this imply that the total parameter count of TGR is at least (E+1)times that of a standard Prompt-DT?
- See Weaknesses.

**Limitations:**

I did not find a discussion of the limitations of the proposed method in the current submission. I would suggest that the authors add a dedicated discussion of limitations, ideally in connection with the questions and weaknesses raised above.

**Strengths And Weaknesses:**

Strengths

- The paper is well-structured, starting from the core issue of unstable expert routing in offline meta-reinforcement learning and logically deriving the design motivation for the TGR.
- The experimental evaluation demonstrates that TGR significantly outperforms existing sota methods.

Weaknesses:

- The paper introduces three distinct nodes for task representation, which significantly increases the total parameter count and the complexity of the training phase. However, the paper lacks a quantitative analysis regarding this additional computational overhead. For instance, a detailed comparison with baselines like MetaDT in terms of parameter efficiency and a discussion on the model's scaling laws would be essential for a comprehensive understanding.
- The SIP algorithm used for state clustering is generally effective in low-dimensional continuous spaces but faces significant scalability challenges in high-dimensional scenarios, such as pixel-level visual inputs, where structured partitioning of the state space becomes extremely difficult. Furthermore, there is a potential risk of "negative transfer": if the state distribution of the test tasks shifts significantly from that of the training set, the Behavioral Trajectory Nodes may provide misleading priors that adversely affect performance rather than aiding generalization.
- The experimental evaluation is primarily confined to relatively basic control tasks, characterized by high inter-task similarity. To provide more persuasive evidence of the framework's robustness, it is highly recommended to conduct experiments in more complex environments with greater task diversity and larger task counts. Benchmark Meta-World [1] provides the ML10 and ML45 task sets.

    [1] Meta-world: A benchmark and evaluation for multi-task and meta reinforcement learning. CoRL. 2020.
- Minor Issues
    - Line 217: The dataset is denoted as $\{\tau_{i,j}\}_{j=1}^N$, implying a total of $N$ trajectories. However, the subsequent random sampling also uses a trajectory count of $N$. It should be clarified whether the model is sampling from the entire set or if this is a notation inconsistency.
    - Figure 2: In the TGR block diagram, the label "expert_i" is somewhat misleading as it refers to the routing component. It is suggested to revise this to "router of expert_i" or a similar descriptor to prevent confusion with the actual "Expert 1 to Expert n" modules shown above.
    - The MoE results presented in Table 2 appear to be a subset of those in Table 4. Streamlining these sections to eliminate overlap would improve the paper's conciseness.

---

> ### Author Rebuttal · Authors · 2026-03-30
>
> ***
>
> **W1&Q2: Computational overhead, parameter efficiency, scaling laws, and the MoE routing mechanism design.**
>
> **Response:** We thank the reviewer for highlighting the need for a quantitative analysis of computational overhead and architectural design.
> First, the three distinct nodes do not all introduce significant trainable parameters. The semantic and dynamics encoders are frozen pre-trained modules. Trajectory nodes are constructed via offline clustering, introducing no additional learned parameters. The parameter increase is strictly localized to the expert and routing modules.
>
> Second, regarding the MoE routing design, each expert is **not** a complete Prompt-DT policy network. We do not duplicate embeddings, self-attention stacks, or output heads. Each expert corresponds only to a block-wise feed-forward network (FFN/MLP) sub-module. The router computes gating weights based on the intermediate representations of the **shared** backbone. The total parameter count is approximately $\theta_{prompt-dt} + E \times \theta_{FFN} + \theta_{moe}$. We strictly avoid an $(E+1)$-fold replication of the entire network.
>
> Third, regarding scaling laws, TGR has 33.7M parameters. Specifically, based on the aforementioned formula, the base Prompt-DT backbone ($\theta_{prompt-dt}$) is 5.5M, each of the $E=9$ FFN experts ($\theta_{FFN}$) is 2.7M, and the MoE router ($\theta_{moe}$) is 3.3M, which yields $5.5\text{M} + 9 \times 2.7\text{M} + 3.3\text{M} = 33.7\text{M}$. To investigate if our gains stem merely from larger capacity, we trained scaled-up baselines (e.g., Meta-DT Large with ~34M parameters).Specifically, we scaled up the Meta-DT backbone by increasing the number of Transformer layers from 6 to 10 and the embedding dimension from 256 to 512.
>
> As shown below, simply scaling up does not yield significant improvements, demonstrating that TGR's performance fundamentally originates from structured task priors and task-guided routing.
>
> | Model | Params | HalfCheetah-Vel | Hopper-Param |
> | :--- | :--- | :--- | :--- |
> | Meta-DT (Large) | ~34M | -97.46 ± 5.28 | 350.47 ± 3.62 |
> | M3DT | ~33M | -107.19 ± 3.97 | 345.49 ± 3.26 |
> | **TGR (Ours)** | **~33M** | **-92.64 ± 6.56** | **370.52 ± 6.54** |
>
> **W2: Scalability of the SIP algorithm in high-dimensional spaces and the potential risk of "negative transfer".**
>
> **Response:**
>
> Regarding scalability: We acknowledge SIP may face representation challenges with high-dimensional image inputs. However, this paper focuses on the state-based offline meta-RL setting, where our conclusions are robustly supported. Integrating representation learning with structured clustering for visual inputs is a promising future direction.
>
> Regarding negative transfer: Empirical evidence shows that SIP-extracted behavioral priors remain highly beneficial even under substantial task discrepancies. In Meta-World benchmarks, which exhibit significant distribution shifts, removing SIP structures led to a noticeable performance drop. This indicates the behavioral trajectory nodes provide invariant structural information that aids generalization, rather than misleading priors. We will carefully state the limitation regarding extreme out-of-distribution scenarios in the revision.
> | Model | ML10 | ML45 |
> | :--- | :--- | :--- |
> | TGR w/o SIP |1095.78±512.33|1061.43±433.27|
> | **TGR** | **1338.07±423.00**|**1179.00±461.00**|
>
>
> **W3: Evaluation in more complex environments.**
>
> **Response:** To provide stronger evidence of robustness, we conducted additional experiments on the challenging Meta-World ML10 and ML45 benchmarks. To rigorously evaluate sample efficiency under a strictly controlled computational budget, we standardized the training steps to 20k for both baselines and TGR.
> As illustrated below, TGR consistently outperforms strong baselines, validating its capability to handle complex, multi-task continuous control environments.
>
> | Model | ML10 | ML45 |
> | :--- | :--- | :--- |
> | M3DT | 267.60 ± 30.32 | 353.06 ± 183.00 |
> | Meta-DT | 923.80 ± 371.00 | 876.90 ± 341.40 |
> | **TGR (Ours)** | **1338.07 ± 423.00** | **1179.00 ± 461.00** |
>
> **Q1: The design of $z^{sem}_{i}$ in Eq. 11.**
>
> **Response:** We apologize for the notational confusion. The two  $z^{sem} _ {i}$   terms are functionally identical. Our intention was to express concatenating the semantic representation $z^{sem}_{i}$ to every sequence of the $<state, action, return>$ tuple. We will revise Eq. 11 to explicitly clarify this process and eliminate ambiguity regarding token occupation.
>
> **Response to Minor Issues:**
> *   **Line 217 Notation:** We will distinguish the total dataset from the sampled subset as follows:
> $\mathcal{D} _ i = \{ \tau_{i,j} \mid j = 1, \dots, N_i \}$
> for the full dataset, and $M$ for the sampled trajectories.
> *   **Figure 2 Label:** We will revise "expert_i" to "router of expert_i" to prevent structural confusion.
> *   **Table 2 & 4 Overlap:** We will streamline these sections to eliminate overlapping results.
>
> ***

---

> > ### Author Rebuttal · Reviewer_aJct · 2026-04-01
> >
> > Thank the authors for their comprehensive rebuttal. The responses have successfully addressed the majority of my concerns; therefore, I will raise my score.
> >
> > However, I have a few remaining suggestions for the final version of the manuscript, which is the primary reason I am not providing a higher score at this stage:
> >
> > - Integration of Limitations: As mentioned in my previous review, please ensure that a formal analysis of the work's limitations is integrated into the main text. This should include, but not be limited to, a discussion on image inputs and performance in out-of-distribution (OOD) scenarios.
> > - Clarification on Scaling Laws (W1): Regarding my comment on scaling laws (W1), I would like to clarify that the intent was not merely to scale Meta-DT parameters to the same magnitude for a one-to-one comparison. Instead, the expectation is to demonstrate the scaling behavior—specifically, how the performance of TGR and key baselines (M3DT, Meta-DT) evolves across different parameter scales. I recognize that the rebuttal period is too short to conduct such extensive experiments; however, I strongly encourage the authors to include these comparative scaling results in the final camera-ready version if the paper is accepted.

---

> > > ### Author Response · Authors · 2026-04-03
> > >
> > > Dear Reviewer aJct,
> > >
> > > Thank you very much for your continued engagement, constructive feedback, and for raising your score. We are glad that our rebuttal has addressed most of your concerns.
> > >
> > > We fully agree with your remaining suggestions and will incorporate them into the camera-ready version if accepted. Specifically, we will add a formal discussion of limitations, clarifying that our current framework is designed for low-dimensional state-based offline meta-RL and does not directly extend to image-based observations without additional challenges in semantic alignment, behavioral clustering, and dynamics modeling. We will also include a more systematic scaling study of TGR and key baselines across a broad parameter range (approximately 2M to 70M, chosen as the smallest and largest practically meaningful scales under the current task setting), together with evaluations on OOD generalization and robustness across different dataset qualities.
> > >
> > > Thank you again for your valuable suggestions.

---

### Official Review · Reviewer_yc9m · 2026-03-12

**Soundness:** 3
**Presentation:** 3
**Significance:** 3
**Originality:** 3
**Overall Recommendation:** 5
**Confidence:** 3

**Summary:**

This work proposes a mixture of experts (MoE) solution to the problem of offline meta-reinforcement learning (OMRL). Concretely, the authors address the insight that, traditionally, OMRL methods treat all tasks uniformly ignoring the fact that some seen tasks might be more related to a given unseen task than others.

To address this, the authors use a MoE approach, letting the expert specialize on a set of similar tasks. To cluster training tasks and select an expert at test time, they introduce three components ("nodes") meant to encode task information. A dynamics model encodes transition dynamics, a pretrained language model provides task semantics, and a behavioral node encodes observation-sequences as behavior without policy-specific biases. The distribution of training tasks is clustered using semantic and Prompt-DT information to train specialized experts.

The authors then introduce the task-guided router (TGR) which is used to select experts based on the task information encoded by the nodes.

**Compliance With Llm Reviewing Policy:**

Affirmed.

**Final Justification:**

The rebuttal comprehensively addressed my concerns and similar concerns by other reviewers.

The experiments showing improvements under equivalent parameter counts and training steps are convincing. The additional experiments showing superior performance in more complex generalization tasks with lower inter-task similarity, as well as augmenting baselines with task metadata, significantly improve the contributions of this paper, which is why I am increasing my score.

**Key Questions For Authors:**

1. How important is tuning of the natural-language task description prompt, how robust is the proposed method to variation?
2. How were the number of experts selected? How robust is the method with respect to this hyperparameter?

**Limitations:**

A discussion of the limitations would be desirable.
- This method requires task descriptions (metadata) at test time.
- Methods and baselines have different training pipelines and parameter counts. Could this maybe be fairly compared based on wall-clock time?

**Strengths And Weaknesses:**

Strengths
- Since OMRL aims to learn a *distribution* of tasks, the reasoning that some unseen tasks are more closely related to some training tasks than others make sense. The idea of encoding higher-level task information, in order to disentangle it from behavior, is shared by prior works (such as Meta-DT). The idea of explicitly training specialized experts is novel and its validity supported by empirical results.
- While MoE has been done for offline mult-task RL< I'm not aware of work on OMRL. Additionally, the paper introduces further improvements (accompanied by ablations) to the basic MoE approach.
- The paper is clearly written and structured and the methods are explained well.
- Ablations clearly demonstrate how the three node components contribute to the overall performance.

Weaknesses
- One of the main motivations for OMRL in general and this paper in particular is performance on unseen tasks. This paper performs experiments in a number of established environments, that have fairly simple generalization, often controlled by a single parameter. This is defensible, as established works in OMLR use the same environments (e.g. Meta-DT), which is why I'm accepting the paper. For me to go from weak accept to accept, I would like to see some enviornments with more challenging generalization, such as MetaWorld ML10 and ML45. These forms of generalization might be a stress test for this paper's approach in particular, since the notion of similarity between seen and unseen tasks is more complex.

---

> ### Author Rebuttal · Authors · 2026-03-30
>
> **W1&Limit1: Generalization in more challenging environments and robustness to prompt variations.**
>
> **Response:** First,to demonstrate TGR's generalization in more challenging and diverse environments,we evaluated it alongside strong baselines on the Meta-World ML10 and ML45 benchmarks.To rigorously evaluate sample efficiency under a strictly controlled computational budget, we standardized the training steps to 20k for both baselines and TGR.
>
> Second,to investigate robustness to prompt tuning,we evaluated TGR using two prompt variations:
> 1. **Complex prompt:** A structured JSON (e.g., `{"description":"Rotate...","env_type":"metaworld","state_dim":39, ...}`).
> 2. **Simple prompt:** Only the description (e.g., `{"description":"Rotate..."`).
>
> It is true that task descriptions are needed during testing,but our task descriptions only require simple construction.Moreover, much of the information in the task descriptions is originally inherent to the tasks themselves,such as task types or status dimensions.The comprehensive results are summarized in the table below:
>
> |Model|ML10|ML45|
> | :--- | :--- | :--- |
> |M3DT|267.60±30.32|353.06±183.00|
> |Meta-DT|923.80±371.00|876.90±341.40|
> |**TGR(complex prompt)**|**1338.07±423.00**|**1179.00±461.00**|
> |TGR(simple prompt)|1289.58±371.56|1089.04±527.46|
>
> 1. **Superior Generalization:** Regardless of the prompt type used,TGR consistently and significantly outperforms the strong baselines (M3DT and Meta-DT), validating its robustness in highly complex settings.Moreover,since M3DT itself is designed for multi-task RL,it performs particularly poorly in complex metaworld meta-learning.
> 2. **Robustness to Prompt Variation:** The empirical results show no significant performance drop when switching from complex to simple prompts. This robust behavior stems from our architectural design.Before feeding metadata into the Transformer, a TaskEmbedding layer ( Eq. 4) secondarily encodes it, which effectively extracts the essential task-specific information
>
> **Q2:how to select the number of experts**
>
> **Response:** In our main experiments, we set the number of experts to 9. This choice aligns with M3DT and was validated by our preliminary experiments. Our rationale is that too few experts limit the model's capacity to distinguish diverse tasks, whereas too many experts dilute the effective training signals assigned to each expert.
> To explicitly evaluate the sensitivity to this hyperparameter, we conducted additional ablation studies on two representative environments (HalfCheetah-Vel and Hopper-Param) using 4, 9, and 15 experts.
>
> |Model|HalfCheetah-Vel|Hopper-Param|
> | :--- | :--- | :--- |
> |Meta-DT|-99.28±3.96 |348.20±3.21|
> |TGR (4 Experts)|-95.34±5.28|362.80±5.49|
> |**TGR (9 Experts)**|**-92.64±6.56**|**370.52±6.54**|
> | TGR (15 Experts) | -98.73±7.34 | 363.60±5.89 |
>
> As indicated, "more experts" is not strictly better. 4 experts restrict task-partitioning, while 15 leads to fragmented data and unstable routing. 9 experts strike an optimal balance. Notably, even with 4 or 15 experts, TGR still outperforms baselines. This aligns with MoE literature (e.g., M3DT), showing that TGR's gains originate from the structured task-guided routing rather than simply scaling up expert counts.
>
> **Limit2: Fair comparison regarding different training pipelines and parameter counts.**
>
> **Response:** While TGR indeed employs a multi-stage approach, the total computational budget is strictly controlled to ensure fairness.Specifically,we sequentially train the Prompt-DT backbone,freeze its parameters to train the individual experts,and finally freeze both to train the MoE router.As detailed in Appendix C and Figure 7,the number of gradient update steps for each stage is strictly identical to the standard training steps used for the baselines (e.g., Meta-DT).Therefore,the comparison remains highly fair in terms of total optimization steps.
>
> Regarding parameter counts, to investigate whether our performance gains stem merely from a larger model capacity, we trained scaled-up versions of the baselines (e.g., Meta-DT Large) to match TGR's capacity (~33M parameters). Specifically, we scaled up the Meta-DT backbone by increasing the number of Transformer layers from 6 to 10 and the embedding dimension from 256 to 512.
>
> This scaling strategy is highly reasonable and standard in Transformer architectures, as it uniformly expands the model's depth (layers) and width (embedding and heads), ensuring that the baseline model has equivalent representational capacity and parameter scale to our MoE-augmented TGR.
>
> |Model|Params|HalfCheetah-Vel|Hopper-Param|
> | :--- | :--- | :--- | :--- |
> |Meta-DT(Large)|~34M|-97.46±5.28|350.47±3.62|
> |M3DT|~33M|-107.19±3.97|345.49±3.26|
> |**TGR (Ours)**|**~33M**|**-92.64±6.56**|**370.52±6.54**|
>
> As shown in the table,simply scaling up the baseline yields no significant improvements.This shows TGR's performance stems from structured priors and routing, rather than parameter expansion.
> ***

---

> > ### Author Rebuttal · Reviewer_yc9m · 2026-04-03
> >
> > I thank the authors for their time in putting together this comprehensive rebuttal.
> >
> > The experiments showing improvements under equivalent parameter counts and training steps are convincing. The additional experiments showing superior performance in more complex generalization tasks with lower inter-task similarity, as well as augmenting baselines with task metadata, significantly improve the contributions of this paper, which is why I am increasing my score.
> >
> > I agree with other reviewers that image-based experiments would further increase the impact of this paper, though this is not necessary for acceptance and is not usually done in OMRL.
> >
> > Good work!

---

> > > ### Author Response · Authors · 2026-04-04
> > >
> > > Dear Reviewer  yc9m
> > >
> > > Thank you very much for your thorough and constructive feedback, and for increasing your score from 4 to 5. We truly appreciate the time and effort you put into reviewing our work.Your positive recognition of our additional experiments  means a lot to us. We are also grateful for your understanding regarding the image-based experiments.
> > >
> > > Thank you again for your support and for helping us improve our paper.

---

### Decision · Program_Chairs · 2026-04-30

**Decision:**

Accept (regular)

**Comment:**

Overall, the reviewers agree that this paper is a solid contribution to offline meta-RL. The method is well-motivated and achieves consistent gains over strong baselines, supported by solid experiments and ablations. The paper is also clear and well-structured, and the rebuttal helped address most of the major concerns.

The main remaining critiques are limited technical novelty and insufficient evaluation on more challenging OOD and visual settings. Overall, the AC recommends acceptance as a solid and well-executed contribution, with clear empirical value and a meaningful direction for future work.